# Splat-SLAM: Globally Optimized RGB-only SLAM with 3D Gaussians

## Abstract

3D Gaussian Splatting has emerged as a powerful representation of geometry and appearance for RGB-only dense Simultaneous Localization and Mapping (SLAM), as it provides a compact dense map representation while enabling efficient and high-quality map rendering. However, existing methods show significantly worse reconstruction quality than competing methods using other 3D representations, *e.g.* neural points clouds, since they either do not employ global map and pose optimization or make use of monocular depth. In response, we propose the first RGB-only SLAM system with a dense 3D Gaussian map representation that utilizes all benefits of globally optimized tracking by adapting dynamically to keyframe pose and depth updates by actively deforming the 3D Gaussian map. Moreover, we find that refining the depth updates in inaccurate areas with a monocular depth estimator further improves the accuracy of the 3D reconstruction. Our experiments on the Replica, TUM-RGBD, and ScanNet datasets indicate the effectiveness of globally optimized 3D Gaussians, as the approach achieves superior or on par performance with existing RGB-only SLAM methods methods in tracking, mapping and rendering accuracy while yielding small map sizes and fast runtimes. The source code will be publicly available.

## 1 Introduction

A common factor within the recent trend of dense SLAM is that the majority of works reconstruct a dense map by optimizing a neural implicit encoding of the scene, either as weights of an MLP Azinović et al. (2022); Sucar et al. (2021); Matsuki et al. (2023b); Ortiz et al. (2022), as features anchored in dense grids Zhu et al. (2022); Newcombe et al. (2011); Weder et al. (2020; 2021); Sun et al. (2021); Božič et al. (2021); Li et al. (2022); Zou et al. (2022); Sandström et al. (2023), using hierarchical octrees Yang et al. (2022a), via voxel hashing Zhang et al. (2023b;a); Chung et al. (2022); Rosinol et al. (2022); Matsuki et al. (2023c), point clouds Hu et al. (2023); Sandström et al. (2023); Liso et al. (2024); Zhang et al. (2024) or axis-aligned feature planes Mahdi Johari et al. (2022); Peng et al. (2020). We have also seen the introduction of 3D Gaussian Splatting (3DGS) to the dense SLAM field Yugay et al. (2023); Keetha et al. (2023); Yan et al. (2023); Matsuki et al. (2023a); Huang et al. (2023).

Out of this 3D representation race there is, however, not yet a clear winner. In the context of dense SLAM, a careful modeling choice needs to be made to achieve accurate surface reconstruction as well as low tracking errors. Some takeaways can be deduced from the literature: neural implicit point cloud representations achieve state-of-the-art reconstruction accuracy Liso et al. (2024); Zhang et al. (2024); Sandström et al. (2023), especially with RGBD input. At the same time, 3D Gaussian splatting methods yield the highest fidelity renderings Matsuki et al. (2023a); Yugay et al. (2023); Keetha et al. (2023); Huang et al. (2023); Yan et al. (2023) and show promise in the RGB-only setting due to their flexibility in optimizing the surface location Huang et al. (2023); Matsuki et al. (2023a). However, they are not leveraging any multi-view depth or geometric prior leading to poor geometry in the RGB-only setting. The majority of the aforementioned works *only* deploy so called frame-to-model tracking, and do not implement global trajectory and map optimization, leading to excessive drift, especially in real world conditions. Instead, to this date, frame-to-frame tracking methods, coupled with loop closure and global bundle adjustment (BA) achieve state-of-the-art tracking accuracy Zhang et al. (2023b;a; 2024). However, they either use hierarchical feature grids Zhang et al. (2023b;a), not suitable for map deformations at *e.g.* loop closure as they require expensive reintegration strategies,

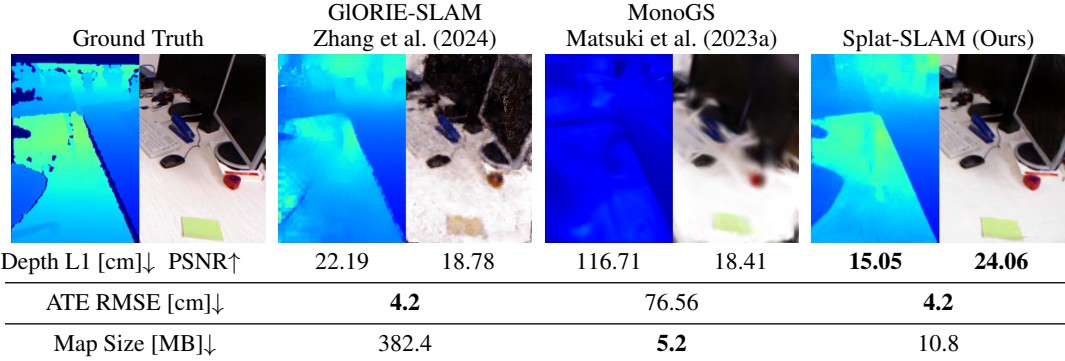

| | | GlORIE-SLAM Zhang et al. (2024) | | MonoGS Matsuki et al. (2023a) | | Splat-SLAM (Ours) | |
|---|---|---|---|---|---|---|---|
| Depth L1 [cm]↓ PSNR↑ | | 22.19 | 18.78 | 116.71 | 18.41 | **15.05** | **24.06** |
| ATE RMSE [cm]↓ | | **4.2** | | 76.56 | | **4.2** | |
| Map Size [MB]↓ | | 382.4 | | **5.2** | | 10.8 | |

Figure 1: **Splat-SLAM.** Our system yields accurate scene reconstruction (rendering depth L1) and rendering (PSNR) and on par tracking accuracy (ATE RMSE) to GlORIE-SLAM and map size to MonoGS. The results averaged over all keyframes. The scene is from TUM-RGBD Sturm et al. (2012) `fr1 room`.

or neural point clouds as in GlORIE-SLAM Zhang et al. (2024). While the neural point cloud is straightforward to deform, the depth guided rendering leads to artifacts when the depth is noisy and the surface estimation can only be adjusted locally since the point locations are not optimized directly.

In this work we propose an RGB-only SLAM system that combines the strengths of frame-to-frame tracking using recurrent dense optical flow Teed & Deng (2021) with the fidelity of 3D Gaussians as the map representation Matsuki et al. (2023a) (see fig. 1). The point-based 3D Gaussian map enables online map deformations at loop closure and global BA. To enable accurate surface reconstruction, we leverage consistent so called proxy depth that combines multi-view depth estimation with learned monocular depth.

Our contribution comprises, for the first time, a SLAM pipeline encompassing all the following parts:

- A frame-to-frame RGB-only tracker with global consistency.
- A dense deformable 3D Gaussian map that adapts online to loop closure and global BA.
- A proxy depth map consisting of on-the-fly optimized multi-view depth and a monocular depth estimator leading to improved rendering and reconstruction quality.
- Improved map sizes and runtimes compared to other dense SLAM approaches.

## 2 RELATED WORK

**Dense Visual SLAM.** Curless and Levoy Curless & Levoy (1996) pioneered dense online 3D mapping with truncated signed distance functions, with KinectFusion Newcombe et al. (2011) demonstrating real-time SLAM via depth maps. Enhancements like voxel hashing Nießner et al. (2013); Kähler et al. (2015); Oleynikova et al. (2017); Dai et al. (2017b); Matsuki et al. (2023c) and octrees Steinbrucker et al. (2013); Yang et al. (2022a); Marniok et al. (2017); Chen et al. (2013); Liu et al. (2020) improved scalability, while point-based SLAM Whelan et al. (2015); Schops et al. (2019); Cao et al. (2018); Kähler et al. (2015); Keller et al. (2013); Cho et al. (2021); Zhang et al. (2020); Sandström et al. (2023); Liso et al. (2024); Zhang et al. (2024) has also been effective. To address pose drift, globally consistent pose estimation and dense mapping techniques have been developed, often dividing the global map into submaps Cao et al. (2018); Dai et al. (2017b); Fioraio et al. (2015); Tang et al. (2023); Matsuki et al. (2023c); Maier et al. (2017); Kähler et al. (2016); Stückler & Behnke (2014); Choi et al. (2015); Kähler et al. (2015); Reijgwart et al. (2019); Henry et al. (2013); Bosse et al. (2003); Maier et al. (2014); Tang et al. (2023); Mao et al. (2023); Liso et al. (2024). Loop detection triggers submap deformation via pose graph optimization Cao et al. (2018); Maier et al. (2017); Tang et al. (2023); Matsuki et al. (2023c); Kähler et al. (2016); Endres et al. (2012); Engel et al. (2014); Kerl et al. (2013); Choi et al. (2015); Henry et al. (2012); Yan et al. (2017); Schops et al. (2019); Reijgwart et al. (2019); Henry et al. (2013); Stückler & Behnke (2014); Wang et al. (2016); Matsuki et al. (2023c); Hu et al. (2023); Mao et al. (2023); Liso et al. (2024). Sometimes global BA is used for refinement Dai et al. (2017b); Schops et al. (2019); Cao et al. (2018); Teed & Deng (2021); Yan et al. (2017); Yang et al. (2022b); Matsuki et al. (2023c); Chung et al. (2022); Tang et al. (2023); Hu et al. (2023). 3D

Figure 2: **Splat-SLAM Architecture.** Given an RGB input stream, we track and map each keyframe, initially estimating poses through local bundle adjustment (BA) using a DSPO (Disparity, Scale and Pose Optimization) layer. This layer integrates pose and depth estimation, enhancing depth with monocular depth. It further refines poses globally via online loop closure and global BA. The proxy depth map merges keyframe depths $\tilde{D}$ from the tracking with monocular depth $D^{mono}$ to fill gaps. Mapping employs a deformable 3D Gaussian map, optimizing its parameters through a re-rendering loss. Notably, the 3D map adjusts for global pose and depth updates before each mapping phase.

Gaussian SLAM with RGBD input has also been shown, but these methods do not consider global consistency via *e.g.* loop closure Yugay et al. (2023); Keetha et al. (2023); Yan et al. (2023). Other approaches to global consistency minimize reprojection errors directly, with DROID-SLAM Teed & Deng (2021) refining dense optical flow and camera poses iteratively, and recent enhancements like GO-SLAM Zhang et al. (2023b), HI-SLAM Zhang et al. (2023a), and GlORIE-SLAM Zhang et al. (2024) optimizing factor graphs for accurate tracking. For a recent survey on NeRF-inspired dense SLAM, see Tosi et al. (2024).

**RGB-only Dense Visual SLAM.** The majority of NeRF inspired dense SLAM works using only RGB cameras do not address the problem of global map consistency or requires expensive reintegration strategies via backpropagation Rosinol et al. (2022); Chung et al. (2022); Li et al. (2023); Zhu et al. (2023); Peng et al. (2024); Zhang et al. (2023b;a); Hua et al. (2023); Naumann et al. (2023); Hua et al. (2024). Instead, the concurrent GlORIE-SLAM Zhang et al. (2024) uses a feature based point cloud which can adapt to global map changes in a straight forward way. However, redundant points are not pruned, leading to large map sizes. Furthermore, the depth guided sampling during rendering leads to rendering artifacts when noise is present in the estimated depth. MonoGS Matsuki et al. (2023a) and Photo-SLAM Huang et al. (2023) pioneered RGB-only SLAM with 3D Gaussians. However, they lack proxy depth which prevents them from achieving high accuracy mapping. MonoGS Matsuki et al. (2023a) also lacks global consistency. Concurrent to our work, MoD-SLAM Zhou et al. (2024) uses an MLP to parameterize the map via a unique reparameterization.

## 3 METHOD

Splat-SLAM is a monocular SLAM system which tracks the camera pose while reconstructing the dense geometry of the scene in an online manner. This is achieved through the following steps: We first track the camera by performing local BA on selected keyframes by fitting them to dense optical flow estimates. The local BA optimizes the camera pose as well as the dense depth of the keyframe. For global consistency, when loop closure is detected, loop BA is performed on an extended graph including the loop nodes and edges (section 3.1). Interleaved with tracking, mapping is done on a progressively growing 3D Gaussian map which deforms online to the keyframe poses and so called proxy depth maps (section 3.2). For an overview of our method, see fig. 2.

### 3.1 TRACKING

To predict the motion of the camera during scene exploration, we use a pretrained recurrent optical flow model Teed & Deng (2020) coupled with a Disparity, Scale and Pose Optimization (DSPO) layer Zhang et al. (2024) to jointly optimize camera poses and per pixel disparities. In the following, we describe this process in detail.

Optimization is done with the Gauss-Newton algorithm over a factor graph $G(V, E)$, where the nodes $V$ store the keyframe pose and disparity, and edges $E$ store the optical flow between keyframes. Odometry keyframe edges are added to $G$ by computing the optical flow to the last added keyframe. If the mean flow is larger than a threshold $\tau \in \mathbb{R}$, the new keyframe is added to $G$. Edges for loop closure and global BA are discussed later. Importantly, the same objective is optimized for local BA, loop closure and global BA, but over factor graphs with different structures.

The DSPO layer consists of two optimization objectives that are optimized alternatively. The first objective, typically termed Dense Bundle Adjustment (DBA) Teed & Deng (2021) optimizes the pose and disparity of the keyframes jointly, eq. (1). Specifically, the objective is optimized over a local graph defined within a sliding window over the current frame.

$$\arg\min_{\omega, d} \sum_{(i,j)\in E} \left\| \tilde{p}_{ij} - K\omega_j^{-1}(\omega_i(1/d_i)K^{-1}[p_i, 1]^T) \right\|_{\Sigma_{ij}}^2 \quad , \tag{1}$$

with $\tilde{p}_{ij} \in \mathbb{R}^{(W\times H\times 2)\times 1}$ being the flattened predicted pixel coordinates when the pixels $p_i \in \mathbb{R}^{(W\times H\times 2)\times 1}$ from keyframe $i$ are projected into keyframe $j$ using optical flow. Further, $K$ is the camera intrinsics, $\omega_j$ and $\omega_i$ the camera-to-world extrinsics for keyframes $j$ and $i$, $d_i$ the disparity of pixel $p_i$ and $\|\cdot\|_{\Sigma_{ij}}$ is the Mahalanobis distance with diagonal weighting matrix $\Sigma_{ij}$. Each weight denotes the confidence of the optical flow prediction for each pixel in $\tilde{p}_{ij}$. For clarity of the presentation, we omit homogeneous coordinates.

The second objective introduces monocular depth $D^{\text{mono}}$ as two additional data terms. The monocular depth $D^{\text{mono}}$ is predicted at runtime by a pretrained relative depth DPT model Eftekhar et al. (2021).

$$\arg\min_{d^h, \theta, \gamma} \sum_{(i,j)\in E} \left\| \tilde{p}_{ij} - K\omega_j^{-1}(\omega_i(1/d_i^h)K^{-1}[p_i, 1]^T) \right\|_{\Sigma_{ij}}^2 \tag{2}$$

$$+\alpha_1 \sum_{i\in V} \left\| d_i^h - (\theta_i(1/D_i^{\text{mono}}) + \gamma_i) \right\|^2 + \alpha_2 \sum_{i\in V} \left\| d_i^l - (\theta_i(1/D_i^{\text{mono}}) + \gamma_i) \right\|^2 \quad .$$

Here, the optimizable parameters are the scales $\theta \in \mathbb{R}$, shifts $\gamma \in \mathbb{R}$ and a subset of the disparities $d^h$ classified as being high error (explained later). This is done since the monocular depth is only deemed useful where the multi-view disparity $d_i$ optimization is inaccurate. Furthermore, $\alpha_1 < \alpha_2$, which is done to ensure that the scales $\theta$ and shifts $\gamma$ are optimized with the preserved low error disparities $d^l$. The scale $\theta_i$ and shift $\gamma_i$ are initialized using least squares fitting

$$\{\theta_i, \gamma_i\} = \arg\min_{\theta, \gamma} \sum_{(u,v)} \left( (\theta(1/D_i^{\text{mono}}) + \gamma) - d_i^l \right)^2 \quad . \tag{3}$$

Equation (1) and eq. (2) are optimized alternatively to avoid the scale ambiguity encountered if $d$, $\theta$, $\gamma$ and $\omega$ are optimized jointly.

Next, we describe how high and low error disparities are classified. For a given disparity map $d_i$ (separated into low and high error parts $\{d_i^l, d_i^h\}$) for frame $i$, we denote the corresponding depth $\tilde{D}_i = 1/d_i$. Pixel correspondences $(u, v)$ and $(\hat{u}, \hat{v})$ between keyframes $i$ and $j$ respectively are established by warping $(u, v)$ into frame $j$ with depth $\tilde{D}_i$ as

$$p_i = \omega_i \tilde{D}_i(u, v)K^{-1}[u, v, 1]^T, \qquad [\hat{u}, \hat{v}, 1]^T \propto K\omega_j^{-1}[p_i, 1]^T \quad . \tag{4}$$

The corresponding 3D point to $(\hat{u}, \hat{v})$ is computed from the depth at $(\hat{u}, \hat{v})$ as

$$p_j = \omega_j \tilde{D}_j(\hat{u}, \hat{v})K^{-1}[\hat{u}, \hat{v}, 1]^T \quad . \tag{5}$$

If the L2 distance between $p_i$ and $p_j$ is smaller than a threshold, the depth $\tilde{D}_i(u, v)$ is consistent between $i$ and $j$. By looping over all keyframes except $i$, the global two-view consistency $n_i$ can be computed for frame $i$ as

$$n_i(u, v) = \sum_{\substack{k\in\text{KFs},\\ k\neq i}} \mathbb{1}\left( \|p_i - p_k\|_2 < \eta \cdot \text{average}(\tilde{D}_i) \right) \quad . \tag{6}$$

Here, $\mathbb{1}(\cdot)$ is the indicator function and $\eta \in \mathbb{R}_{\geq 0}$ is a hyperparameter and $n_i$ is the total two-view consistency for pixel $(u, v)$ in keyframe $i$. $\tilde{D}_i(u, v)$ is valid if $n_i$ is larger than a threshold.

**Loop Closure.** To mitigate scale and pose drift, we incorporate loop closure along with online global bundle adjustment (BA) in addition to local window frame tracking. Loop detection is achieved by calculating the mean optical flow magnitude between the current active keyframes (within the local window) and all previous keyframes. Two criteria are evaluated for each keyframe pair: First, the optical flow must be below a specified threshold $\tau_{\text{loop}}$, ensuring sufficient co-visibility between the

views. Second, the time interval between the frames must exceed a predefined threshold $\tau_t$ to prevent the introduction of redundant edges into the graph. When both criteria are met, a unidirectional edge is added to the graph. During the loop closure optimization process, only the active keyframes and their connected loop nodes are optimized to keep the computational load manageable.

**Global BA.** For the online global BA, a separate graph that includes all keyframes up to the present is constructed. Edges are introduced based on the temporal and spatial relationships between the keyframes, as outlined in Zhang et al. (2023b). Following the approach detailed in Zhang et al. (2024), we execute online global BA every 20 keyframes. To maintain numerical stability, the scales of the disparities and poses are normalized prior to each global BA optimization. This normalization involves calculating the average disparity $\bar{d}$ across all keyframes and then adjusting the disparity to $d_{norm} = d/\bar{d}$ and the pose translation to $t_{norm} = \bar{d}t$.

## 3.2 DEFORMABLE 3D GAUSSIAN SCENE REPRESENTATION

We adopt a 3D Gaussian Splatting representation Kerbl et al. (2023) which deforms under DSPO or loop closure optimizations to achieve global consistency. Thus, the scene is represented by a set $\mathcal{G} = \{g_i\}_{i=1}^N$ of 3D Gaussians. Each Gaussian primitive $g_i$, is parameterized by a covariance matrix $\Sigma_i \in \mathbb{R}^{3 \times 3}$, a mean $\boldsymbol{\mu}_i \in \mathbb{R}^3$, opacity $o_i \in [0, 1]$, and color $\mathbf{c}_i \in \mathbb{R}^3$. All attributes of each Gaussian are optimized through back-propagation. The density function of a single Gaussian is described as

$$g_i(\mathbf{x}) = \exp\left(-\frac{1}{2}(\mathbf{x} - \boldsymbol{\mu}_i)^\top \Sigma_i^{-1}(\mathbf{x} - \boldsymbol{\mu}_i)\right) . \tag{7}$$

Here, the spatial covariance $\Sigma_i$ defines an ellipsoid and is decomposed as $\Sigma_i = R_i S_i S_i^T R_i^T$, where $S_i = \text{diag}(s_i) \in \mathbb{R}^{3 \times 3}$ is the spatial scale and $R_i \in \mathbb{R}^{3 \times 3}$ represents the rotation.

**Rendering.** Rendering color and depth from $\mathcal{G}$, given a camera pose, involves first projecting (known as "splatting") 3D Gaussians onto the 2D image plane. This is done by projecting the covariance matrix $\Sigma$ and mean $\boldsymbol{\mu}$ as $\Sigma' = JR\Sigma R^T J^T$ and $\boldsymbol{\mu}' = K\omega^{-1}\boldsymbol{\mu}$, where $R$ is the rotation component of world-to-camera extrinsics $\omega^{-1}$ and $J$ is the Jacobian of the affine approximation of the projective transformation Zwicker et al. (2001). The final pixel color $C$ and depth $D^r$ at pixel $\mathbf{x}'$ is computed by blending 3D Gaussian splats that overlap at a given pixel, sorted by their depth as

$$C = \sum_{i \in \mathcal{N}} \mathbf{c}_i \alpha_i \prod_{j=1}^{i-1}(1 - \alpha_j) \qquad D^r = \sum_{i \in \mathcal{N}} \hat{d}_i \alpha_i \prod_{j=1}^{i-1}(1 - \alpha_j) , \tag{8}$$

where $\hat{d}_i$ is the z-axis depth of the center of the $i$-th 3D Gaussian and the final opacity $\alpha_i$ is the product of the opacity $o_i$ and the 2D Gaussian density as

$$\alpha_i = o_i \exp\left(-\frac{1}{2}(\mathbf{x}' - \boldsymbol{\mu}_i')^\top \Sigma_i'^{-1}(\mathbf{x}' - \boldsymbol{\mu}_i')\right) . \tag{9}$$

**Map Initialization.** For every new keyframe, we adopt the RGBD strategy of MonoGS Matsuki et al. (2023a) for adding new Gaussians to the unexplored scene space. As we do not have access to a depth sensor, we construct a proxy depth map $D$ by combining the inlier multi-view depth $\tilde{D}$ and the monocular depth $D^{\text{mono}}$ as

$$D(u, v) = \begin{cases} \tilde{D}(u, v) & \text{if } \tilde{D}(u, v) \text{ is valid} \\ \theta D^{\text{mono}}(u, v) + \gamma & \text{otherwise} \end{cases} \tag{10}$$

Here, $\theta$ and $\gamma$ are computed as in eq. (3) but using depth instead of disparity.

**Keyframe Selection and Optimization.** Apart from the keyframe selection based on a mean optical flow threshold $\tau$, we additionally adopt the keyframe selection strategy from Matsuki et al. (2023a) to avoid mapping redundant frames.

To optimize the 3D Gaussian parameters, we batch the parameter updates to a local window similar to Matsuki et al. (2023a) and apply a photometric and geometric loss to the proxy depth as well as a scale regularizer to avoid artifacts from elongated Gaussians. Inspired by Matsuki et al. (2023a), we further use exposure compensation by optimizing an affine transformation for each keyframe. The final loss is

$$\min_{\mathcal{G}, \mathbf{a}, \mathbf{b}} \sum_{k \in \text{KFs}} \frac{\lambda}{N_k} |(a_k C_k + b_k) - C_k^{gt}|_1 + \frac{1 - \lambda}{N_k} |D_k^r - D_k|_1 + \frac{\lambda_{reg}}{|\mathcal{G}|} \sum_i^{|\mathcal{G}|} |s_i - \tilde{s}_i|_1 , \tag{11}$$

where KFs contains the set of keyframes in the local window, $N_k$ is the number of pixels per keyframe, $\lambda$ and $\lambda_{reg}$ are hyperparameters, $\mathbf{a} = \{a_1, \ldots, a_k, \ldots\}$ and $\mathbf{b} = \{b_1, \ldots, b_k, \ldots\}$ are the parameters for the exposure compensation and $\tilde{s}$ is the mean scaling, repeated over the three dimensions.

**Map Deformation.** Since our tracking framework is globally consistent, changes in the keyframe poses and proxy depth maps need to be accounted for in the 3D Gaussian map by a non-rigid deformation. Though the Gaussian means are directly optimized, one could in theory let the optimizer deform the map as refined poses and proxy depth maps are provided. We find, however, that in particular rendering is aided by actively deforming the 3D Gaussian map. We apply the deformation to all Gaussians which receive updated poses and depths before mapping.

Each Gaussian $g_i$ is associated with a keyframe that anchored it to the map $\mathcal{G}$. Assume that a keyframe with camera-to-world pose $\omega$ and proxy depth $D$ is updated such that $\omega \rightarrow \omega'$ and $D \rightarrow D'$. We update the mean, scale and rotation of all Gaussians $g_i$ associated with the keyframe. Association is determined by what keyframe added the Gaussian to the scene. The mean $\boldsymbol{\mu}_i$ is projected into $\omega$ to find the pixel correspondence $(u, v)$. Since the Gaussians are not necessarily anchored on the surface, instead of re-anchoring the mean at $D'$, we opt to shift the mean by $D'(u, v) - D(u, v)$ along the optical axis. We update $R_i$ and $s_i$ accordingly as

$$\boldsymbol{\mu}_i' = \left(1 + \frac{D'(u, v) - D(u, v)}{(\omega^{-1}\boldsymbol{\mu}_i)_z}\right)\omega'\omega^{-1}\boldsymbol{\mu}_i \ , R_i' = R'R^{-1}R_i \ , s_i' = \left(1 + \frac{D'(u, v) - D(u, v)}{(\omega^{-1}\boldsymbol{\mu}_i)_z}\right)s_i \ . \tag{12}$$

Here, $(\cdot)_z$ denotes the z-axis depth. For Gaussians which project into pixels with missing depth or outside the viewing frustum, we *only* rigidly deform them. After the final global BA optimization, we additionally deform the Gaussian map and perform a set of final refinements (see suppl. material).

## 4 EXPERIMENTS

We first describe our experimental setup and then evaluate our method against state-of-the-art dense RGB and RGBD SLAM methods on Replica Straub et al. (2019) as well as the real world TUM-RGBD Sturm et al. (2012) and the ScanNet Dai et al. (2017a) datasets. For more experiments and details, we refer to the supplementary material.

**Implementation Details.** For the proxy depth, we use $\eta = 0.01$ to filter points and use the condition $n_c \geq 2$ to ensure multi-view consistency. For the mapping loss function, we use $\lambda = 0.8$, $\lambda_{reg} = 10.0$. We use 60 iterations during mapping. For tracking, we use $\alpha_1 = 0.01$ and $\alpha_2 = 0.1$ as weights for the DSPO layer. We use the flow threshold $\tau = 4.0$ on ScanNet, $\tau = 3.0$ on TUM-RGBD and $\tau = 2.25$ on Replica. The threshold for loop detection is $\tau_{\text{loop}} = 25.0$. The time interval threshold is $\tau_t = 20$. We conducted the experiments on a cluster with an NVIDIA A100 GPU.

**Evaluation Metrics.** For rendering we report PSNR, SSIM Wang et al. (2004) and LPIPS Zhang et al. (2018) on the rendered keyframe images against the sensor images. For reconstruction, we first extract the meshes with marching cubes Lorensen & Cline (1987) as in Sandström et al. (2023) and evaluate the meshes using accuracy $[cm]$, completion $[cm]$ and completion ratio $[\%]$ (threshold 5 cm) against the ground truth meshes. We also report the re-rendering depth L1 $[cm]$ metric to the ground truth sensor depth as in Rosinol et al. (2022). We use ATE RMSE $[cm]$ Sturm et al. (2012) to evaluate the estimated trajectory.

**Datasets.** We use the RGBD trajectories from Sucar et al. (2021) captured from the synthetic Replica dataset Straub et al. (2019). We also test on real-world data using the TUM-RGBD Sturm et al. (2012) and the ScanNet Dai et al. (2017a) datasets.

**Baseline Methods.** We compare our method to numerous published and concurrent works on dense RGB and RGBD SLAM. Concurrent works are denoted with an asterix[*]. The main baselines are GlORIE-SLAM Zhang et al. (2024) and MonoGS Matsuki et al. (2023a).

**Rendering.** In tab. 1, we evaluate the rendering performance on Replica Straub et al. (2019) and find that our method performs superior among all baseline RGB-methods. Tab. 2 and tab. 3 show the rendering accuracy on the ScanNet Dai et al. (2017a) and TUM-RGBD Sturm et al. (2012) datasets. In particular, we outperform existing RGB-only works with a clear margin, while even beating the currently best RGBD method, Gaussian-SLAM Yugay et al. (2023) on most metrics, despite the fact that we do not implement view-dependent rendering in the form of spherical harmonics.

| Metric | GO-SLAM (Zhang et al., 2023b) | NICER-SLAM (Zhu et al., 2023) | MoD-SLAM* (Li et al., 2023) | Photo-SLAM (Huang et al., 2023) | Mono-GS (Matsuki et al., 2023a) | GlORIE-SLAM* (Zhang et al., 2024) | Q-SLAM* (Peng et al., 2024) | **Ours** |
|---|---|---|---|---|---|---|---|---|
| PSNR ↑ | 22.13 | 25.41 | 27.31 | 33.30 | 31.22 | 31.04 | 32.49 | **36.45** |
| SSIM ↑ | 0.73 | 0.83 | 0.85 | 0.93 | 0.91 | 0.91 | 0.89 | **0.95** |
| LPIPS ↓ | - | 0.19 | - | - | 0.21 | 0.12 | 0.17 | **0.06** |
| ATE RMSE ↓ | 0.39 | 1.88 | **0.35** | 1.09 | 14.54 | **0.35** | - | **0.35** |

Table 1: **Rendering and Tracking Results on Replica Straub et al. (2019) for RGB-Methods.** Our method outperforms all methods on rendering and performs on par for tracking accuracy. Results are from Tosi et al. (2024) except ours (average over 8 scenes). Best results are highlighted as first, second, third.

| Method | Metric | 0000 | 0059 | 0106 | 0169 | 0181 | 0207 | Avg. |
|---|---|---|---|---|---|---|---|---|
| *RGB-D Input* | | | | | | | | |
| SplaTaM Keetha et al. (2023) | PSNR↑ | 19.33 | 19.27 | 17.73 | 21.97 | 16.76 | 19.80 | 19.14 |
| | SSIM ↑ | 0.66 | 0.79 | 0.69 | 0.78 | 0.68 | 0.70 | 0.72 |
| | LPIPS↓ | 0.44 | 0.29 | 0.38 | 0.28 | 0.42 | 0.34 | 0.36 |
| MonoGS Matsuki et al. (2023a) | PSNR↑ | 18.70 | 20.91 | 19.84 | 22.16 | 22.01 | 18.90 | 20.42 |
| | SSIM ↑ | 0.71 | 0.79 | 0.81 | 0.78 | 0.82 | 0.75 | 0.78 |
| | LPIPS↓ | 0.48 | 0.32 | 0.32 | 0.34 | 0.42 | 0.41 | 0.38 |
| Gaussian-SLAM Yugay et al. (2023) | PSNR↑ | 28.54 | 26.21 | 26.26 | 28.60 | 27.79 | 28.63 | 27.67 |
| | SSIM ↑ | 0.93 | 0.93 | 0.93 | 0.92 | 0.92 | 0.91 | 0.92 |
| | LPIPS↓ | 0.27 | 0.21 | 0.22 | 0.23 | 0.28 | 0.29 | 0.25 |
| *RGB Input* | | | | | | | | |
| GO-SLAM Zhang et al. (2023b) | PSNR↑ | 15.74 | 13.15 | 14.58 | 14.49 | 15.72 | 15.37 | 14.84 |
| | SSIM ↑ | 0.42 | 0.32 | 0.46 | 0.42 | 0.53 | 0.39 | 0.42 |
| | LPIPS↓ | 0.61 | 0.60 | 0.59 | 0.57 | 0.62 | 0.60 | 0.60 |
| MonoGS Matsuki et al. (2023a) | PSNR↑ | 16.91 | 19.15 | 18.57 | 20.21 | 19.51 | 18.37 | 18.79 |
| | SSIM ↑ | 0.62 | 0.69 | 0.74 | 0.74 | 0.75 | 0.70 | 0.71 |
| | LPIPS↓ | 0.70 | 0.51 | 0.55 | 0.54 | 0.63 | 0.58 | 0.59 |
| GlORIE-SLAM* Zhang et al. (2024) | PSNR↑ | 23.42 | 20.66 | 20.41 | 25.23 | 21.28 | 23.68 | 22.45 |
| | SSIM ↑ | 0.87 | 0.87 | 0.83 | 0.84 | 0.91 | 0.76 | 0.85 |
| | LPIPS↓ | 0.26 | 0.31 | 0.31 | 0.21 | 0.44 | 0.29 | 0.30 |
| **Splat-SLAM (Ours)** | PSNR↑ | 28.68 | 27.69 | 27.70 | 31.14 | 31.15 | 30.49 | 29.48 |
| | SSIM ↑ | 0.83 | 0.87 | 0.86 | 0.87 | 0.84 | 0.84 | 0.85 |
| | LPIPS ↓ | 0.19 | 0.15 | 0.18 | 0.15 | 0.23 | 0.19 | 0.18 |

Table 2: **Rendering Performance on ScanNet Dai et al. (2017a).** Our method performs even better or on par with all RGB-D methods. We take the numbers for SplaTaM and Gaussian-SLAM from Yugay et al. (2023).

We attribute this to our deformable 3D Gaussian map, optimized with strong proxy depth along a globally consistent tracking backend. In fig. 3 and fig. 1 we show renderings on the real-world ScanNet Dai et al. (2017a) and TUM-RGBD Sturm et al. (2012) datasets. Due to high tracking errors, MonoGS Matsuki et al. (2023a) performs poorly on some scenes, yet fails to achieve the same fidelity as our method when the tracking error is low, as a result of the weak geometric constraints during optimization. Our method avoids the artifacts produced by GlORIE-SLAM Zhang et al. (2024) and yields high quality renderings.

**Reconstruction.** We show quantitative and qualitative results on the Replica Straub et al. (2019) dataset in tab. 4 and fig. 4 respectively. Our method achieves the best performance on all metrics. Qualitatively, we show normal shaded meshes from different viewpoints. Our method can reconstruct finer details than existing works, especially around thin structures (*e.g.* second row), where our strong proxy depth coupled with the 3D Gaussian map representation yields superior depth rendering, which directly influences the mesh quality. In contrast, *e.g.* GlORIE-SLAM Zhang et al. (2024) uses depth guided volume rendering, which is sensitive to input depth noise, resulting in inconsistent depth rendering with floating artifacts. MonoGS Matsuki et al. (2023a) suffers significantly from the lack of proxy depth, visible in all scenes. Fig. 1 shows depth rendering on the real-world TUM-RGBD Sturm et al. (2012) `room` scene. We compute the average depth L1 error over all keyframes, achieving 15.05 cm, beating existing works.

**Ablation Study.** In tab. 5, we conduct a set of ablation studies related to our method, by enabling and disabling certain parts. We find that the combination of filtered multiview depth completed with monocular depth yields the best performance in terms of rendering and reconstruction metrics.

| Method | Method | f1/desk | f2/xyz | f3/off | f1/desk2 | f1/room | **Avg.** |
|---|---|---|---|---|---|---|---|
| *RGB-D Input* | | | | | | | |
| SplaTaM Keetha et al. (2023) | PSNR↑ | 22.00 | 24.50 | 21.90 | - | - | - |
| | SSIM↑ | 0.86 | 0.95 | 0.88 | - | - | - |
| | LPIPS↓ | 0.23 | 0.10 | 0.20 | - | - | - |
| Gaussian-SLAM Yugay et al. (2023) | PSNR↑ | 24.01 | 25.02 | **26.13** | 23.15 | 22.98 | 24.26 |
| | SSIM↑ | **0.92** | 0.92 | **0.94** | 0.91 | **0.89** | **0.92** |
| | LPIPS↓ | **0.18** | 0.19 | **0.14** | **0.20** | **0.24** | **0.19** |
| *RGB Input* | | | | | | | |
| Photo-SLAM Huang et al. (2023) | PSNR↑ | 20.97 | 21.07 | 19.59 | - | - | - |
| | SSIM↑ | 0.74 | 0.73 | 0.69 | - | - | - |
| | LPIPS↓ | 0.23 | 0.17 | 0.24 | - | - | - |
| MonoGS Matsuki et al. (2023a) | PSNR↑ | 19.67 | 16.17 | 20.63 | 19.16 | 18.41 | 18.81 |
| | SSIM↑ | 0.73 | 0.72 | 0.77 | 0.66 | 0.64 | 0.70 |
| | LPIPS↓ | 0.33 | 0.31 | 0.34 | 0.48 | 0.51 | 0.39 |
| GlORIE-SLAM* Zhang et al. (2024) | PSNR↑ | 20.26 | 25.62 | 21.21 | 19.09 | 18.78 | 20.99 |
| | SSIM↑ | 0.79 | 0.72 | 0.72 | **0.92** | 0.73 | 0.77 |
| | LPIPS↓ | 0.31 | 0.09 | 0.32 | 0.38 | 0.38 | 0.30 |
| **Splat-SLAM (Ours)** | PSNR↑ | **25.61** | **29.53** | 26.05 | **23.98** | **24.06** | **25.85** |
| | SSIM↑ | 0.84 | 0.90 | 0.84 | 0.81 | 0.80 | 0.84 |
| | LPIPS↓ | **0.18** | **0.08** | 0.20 | 0.23 | **0.24** | **0.19** |

Table 3: **Rendering Performance on TUM-RGBD Sturm et al. (2012).** Our method performs competitively or better than RGB-D methods. For all RGB-D methods, we take the numbers from Yugay et al. (2023).

| Metrics | NeRF-SLAM (Tosi et al., 2024) | DIM-SLAM (Li et al., 2023) | GO-SLAM (Zhang et al., 2023b) | NICER-SLAM (Zhu et al., 2023) | HI-SLAM (Zhang et al., 2023a) | MoD-SLAM* (Zhou et al., 2024) | GlORIE-SLAM* (Zhang et al., 2024) | Mono-GS (Matsuki et al., 2023a) | Q-SLAM* (Peng et al., 2024) | **Ours** |
|---|---|---|---|---|---|---|---|---|---|---|
| Render Depth L1↓ | 4.49 | - | - | - | - | - | - | 27.24 | 2.76 | **2.41** |
| Accuracy↓ | - | 4.03 | 3.81 | 3.65 | 3.62 | 2.48 | 2.96 | 30.61 | - | **2.43** |
| Completion↓ | - | 4.20 | 4.79 | 4.16 | 4.59 | - | 3.95 | 12.19 | - | **3.64** |
| Comp. Rat.↑ | - | 79.60 | 78.00 | 79.37 | 80.60 | - | 83.72 | 40.53 | - | **84.69** |

Table 4: **Reconstruction Results on Replica Straub et al. (2019) for RGB-Methods.** Our method outperforms existing works on all metrics. Results are averaged over 8 scenes.

**Memory and Runtime.** In tab. 6, we evaluate the peak GPU memory usage, map size and runtime of our method. We achieve a comparable GPU memory usage with GO-SLAM Zhang et al. (2023b) and SplaTaM Keetha et al. (2023). Our map size is similar to MonoGS Matsuki et al. (2023a) and much smaller than GlORIE-SLAM, which does not prune redundant neural points. In fig. 1 we also show similar map size to MonoGS on the real-world TUM-RGBD Sturm et al. (2012) room scene.

| Mono Depth | Multiview Depth | Multiview Filtering | PSNR [dB]↑ | Acc. [cm]↓ | Comp. [cm]↓ | Comp. Ratio [cm]↑ |
|---|---|---|---|---|---|---|
| ✓ | ✗ | ✗ | 36.02 | 3.62 | 4.08 | 81.16 |
| ✗ | ✓ | ✓ | 36.17 | 2.64 | 4.73 | 80.12 |
| ✗ | ✓ | ✗ | 36.21 | 18.71 | 4.06 | 80.29 |
| ✓ | ✓ | ✓ | **36.45** | **2.43** | **3.64** | **84.69** |

Table 5: **Ablation Study on Replica Straub et al. (2019).** We show that the combination of filtered multiview depth completed with monocular depth yields the best performance on all metrics. Mono Depth refers to $D^{mono}$, Multiview Depth refers to $\tilde{D}$ and Multiview Filtering means enabling eq. (6). All results are averaged over 8 scenes.

| | GO-SLAM Zhang et al. (2023b) | SplaTAM Keetha et al. (2023) | GlORIE-SLAM* Zhang et al. (2024) | MonoGS Matsuki et al. (2023a) | Ours |
|---|---|---|---|---|---|
| GPU Usage [GiB] | 18.50 | 18.54 | 15.22 | **14.62** | 17.57 |
| Map Size [MB] | - | - | 114.0 | 6.8 | **6.5** |
| Avg. FPS | **8.36** | 0.14 | 0.23 | 0.32 | 1.24 |

Table 6: **Memory and Running Time Evaluation on Replica Straub et al. (2019) `room0`.** Our peak memory usage and runtime are comparable to existing works. We take the numbers from Tosi et al. (2024) except for ours and MonoGS and we add the Map Size, which denotes the size of the final 3D representation. GPU Usage denotes the peak usage during runtime. All methods are evaluated on an NVIDIA RTX 3090 GPU using single threading for fairness.

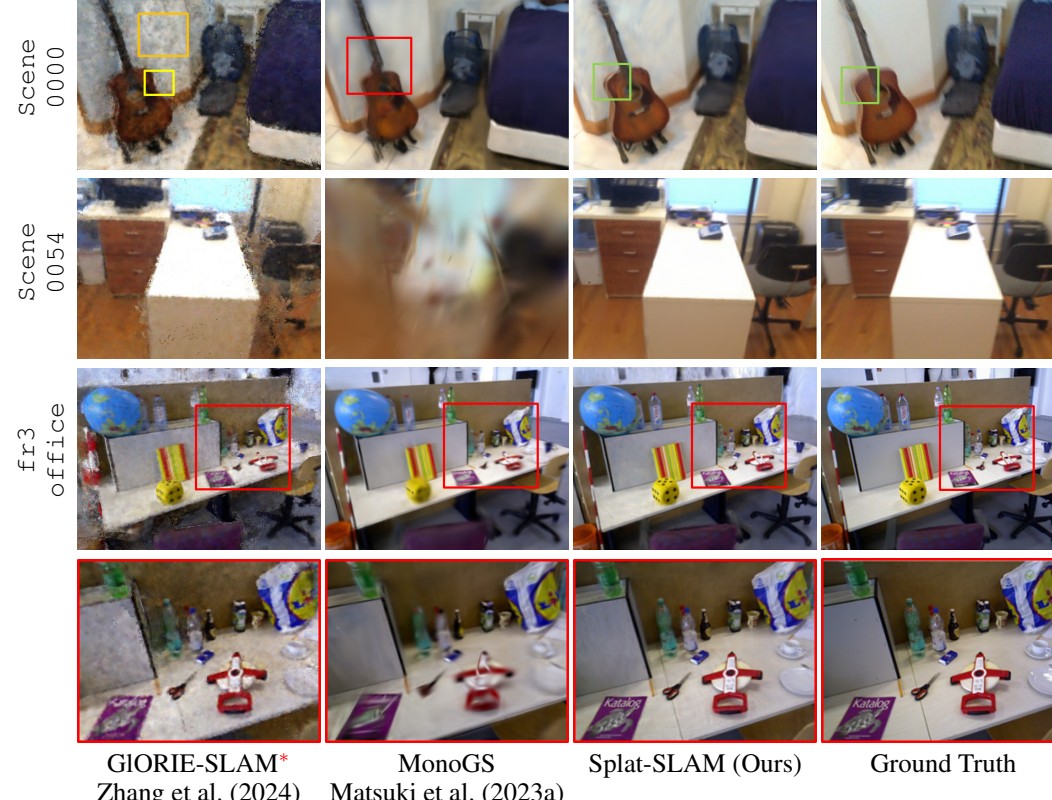

GlORIE-SLAM*      MonoGS      Splat-SLAM (Ours)      Ground Truth
Zhang et al. (2024)    Matsuki et al. (2023a)

Figure 3: **Rendering Results on ScanNet Dai et al. (2017a) and TUM-RGBD Sturm et al. (2012).** Our method yields better rendering quality than GlORIE-SLAM and MonoGS. Top row: the orange box shows artifacts from GlORIE-SLAM, partly due to the depth guided volume rendering. The yellow box shows an area with redundant floating points. The red box shows a rendering distortion, likely from the large trajectory error. The green boxes show that our method fuses information from multiple views to avoid motion blur, present in the input. Fourth row: The rendering is from the pose of the red box in the third row.

Regarding runtime, we are faster than SplaTaM and GlORIE-SLAM and comparable to MonoGS. GO-SLAM has the fastest runtime, but as shown in tab. 1 and tab. 4, it sacrifices rendering and reconstruction quality for speed.

**Limitations.** We currently do not model the appearance with spherical harmonics, since it only yields a marginal gains in rendering accuracy, while requiring more memory. It is is straightforward to add. We only make use of globally optimized frame-to-frame tracking, which fails to leverage frame-to-model queues from the 3D Gaussian map. Another limitation is that our construction of the final proxy depth $D$ is quite simple and does not fuse the monocular and keyframe depths in an informed manner, *e.g.* using normal consistency. Finally, as future work, it is interesting to study how surface regularization can be enforced via *e.g.* quadric surface elements as in Peng et al. (2024).

## 5   CONCLUSION

We proposed Splat-SLAM, a dense RGB-only SLAM system which uses a deformable 3D Gaussian map for mapping and globally optimized frame-to-frame tracking via optical flow. Importantly, the inclusion of monocular depth into the tracking loop, to refine the scale and to correct the erroneous keyframe depth predictions, leads to better rendering and mapping. By using the monocular depth for completion, mapping is further improved. Our experiments demonstrate that Splat-SLAM outperforms existing solutions regarding reconstruction and rendering accuracy while being on par or better with respect to tracking as well as runtime and memory usage.

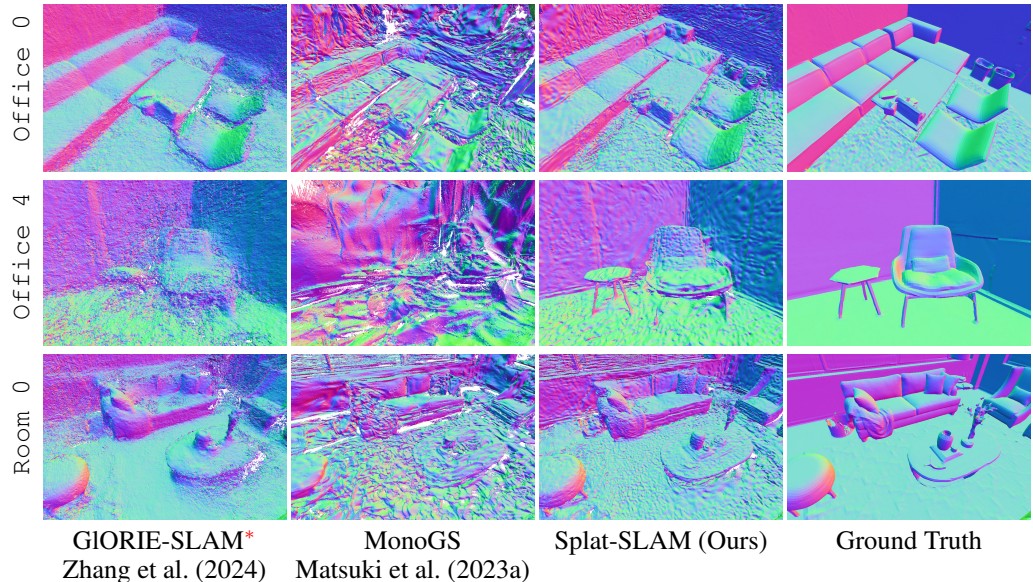

Figure 4: **Reconstruction Results on Replica Straub et al. (2019) on Normal Shaded Meshes.**
Our method achieves higher geometric accuracy compared to existing works. In particular, GlORIE-
SLAM suffers from floating point artifacts (*e.g.* second row) where our method reconstructs even
the individual legs of the table. MonoGS suffers significantly from a lack of proxy depth, despite
multiview optimization.

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
