# OpenReview forum: "Splat-SLAM: Globally Optimized RGB-only SLAM with 3D Gaussians"
_ICLR.cc/2025/Conference — ICLR 2025 Conference Withdrawn Submission_

### Official Review · Reviewer_m1bQ · 2024-10-17

**Soundness:** 3
**Presentation:** 3
**Contribution:** 2
**Rating:** 6
**Confidence:** 4

**Summary:**

This paper addresses the challenges of existing methods of worse reconstruction quality due to the lacing of the global map and pose optimization or monocular depth utilization. The author proposes a dense RGB-only SLAM system with 3D Gaussian representation and utilizes global optimized tracking by actively deforming the 3D Gaussian map. Experiments on the Replica, TUM-RGBD, and ScanNet datasets demonstrate the effectiveness of the proposed RGB-only SLAM methods.

**Strengths:**

The proposed SLAM system combines the strengths of frame-to-frame tracking using recurrent dense optical flow with the fidelity of 3D Gaussians as the map representation without the dependence of depth inputs.
The proposed SLAM system performs better than existing RGB-only SLAM methods in tracking, mapping, and rendering accuracy, and more importantly, yielding small map sizes and fast runtimes.
The paper is well-written, concise, and has excellent formatting of figures and formulas.

**Weaknesses:**

1. The author suggests that this paper proposed the first RGB-only SLAM system with a dense 3D Gaussian map that globally optimized tracking by adapting dynamically to keyframe pose and depth updates. Nevertheless, I argue that this claim is not entirely accurate, since certain prior studies, notably PhotoSLAM, support monocular video as well.

2. Notably, the proposed method falls short of achieving real-time performance, with a frame rate of only 1.24 FPS on the simplest Replica room0 dataset (Table 6). This limitation will hinder the applicability of this method to large-scale, real-world environments. This might related to the depth estimation module for proxy depth prediction.

3. Upon reviewing the paper, I believe that it fails to demonstrate sufficient innovation in the context of 3DGS scene representation, as many of the techniques, e.g., loop closure, global BA., and proxy depth map optimization, have been previously explored in existing literature[1][2][3].

4. Despite the method's claim of not requiring depth input, the algorithm's underlying architecture still employs a depth estimator to compute depth, which are subsequently used to calculate the depth rendering loss and inform pose and map optimization. As a result, the proposed method still fundamentally relies on depth estimation, which in turn contributes to slower online SLAM algorithm performance.

5. Although the paper's experimental design is thorough, it still exhibits some limitations. Primarily, for SLAM algorithms, the accuracy of tracking and the quality of the generated map are of paramount importance. Consequently, rendering quality is somewhat secondary. The authors should prioritize including more figures illustrating tracking accuracy in the main text, rather than relegating them to the supplementary materials. Additionally, I highly recommend that the authors incorporate a comparison with PhotoSLAM to further bolster the experimental setup.

[1] Photo-SLAM: Real-time Simultaneous Localization and Photorealistic Mapping
for Monocular, Stereo, and RGB-D Cameras
[2] plaTAM: Splat, Track & Map 3D Gaussians for Dense RGB-D SLAM
[3] NICER-SLAM Neural Implicit Scene Encoding for RGB SLAM
[4] Deformable 3D Gaussians for High-Fidelity Monocular Dynamic Scene Reconstruction

**Questions:**

1. According to Table 6, Splat-SLAM achieves superior memory usage and runtime. I noticed the running time in Tab.6, can author provide a time breakdown of each component? How many 3dgs point does the model use?

2. Table 6 shows that SPLAT-SLAM requires a mere 6.5MB of storage, which is equivalent to a substantial 135,416 Gaussian points (6.5 / 4 / (3 + 3 + 4 + 1 + 1) * 10^6) = 135416 Gaussians, is nothing short of remarkable. Unfortunately, the lack of a demonstration video to substantiate this claim is a significant omission.

3. In Section 3.2, the authors introduce a deformable 3D Gaussian scene representation[4], which may be perceived as misleading. Deformable Gaussians are conventionally employed in 4D reconstruction, whereas this section elaborates on the optimization process of 3DGS maps subject to loss function constraints via DSPO or loop closure optimizations. The terminology 'deformable 3D Gaussian' is not precisely applicable in this context. I would appreciate it if the authors could provide clarification on this matter.

I have assigned a borderline rating to this manuscript. However, I am more than willing to engage in a constructive discussion and revisit my evaluation in light of the authors' responses to my comments

---

### Official Review · Reviewer_4ssA · 2024-10-27

**Soundness:** 3
**Presentation:** 3
**Contribution:** 3
**Rating:** 6
**Confidence:** 3

**Summary:**

This paper presents an RGB-only 3DGS-based SLAM system addressing key limitations in existing 3DGS-based SLAM approaches, e.g.,  challenges in achieving global map optimization and accurate 3D reconstructions. The proposed method, Splat-SLAM, supports map deformations upon loop closure and incorporates global bundle adjustment (BA) while leveraging a monocular depth estimation model to enhance reconstruction accuracy.

**Strengths:**

1）	This paper is the first RGB-only 3DGS-based SLAM system with loop closure, proxy depth, and online 3D Gaussian map deformations with improved map sizes and runtimes. In my opinion, this is a relatively comprehensive work in the field of 3DGS-based SLAM. As loop closure is a crucial challenge in SLAM, this work enables map deformations at loop closure and integrates global bundle adjustment.

2）	Extensive evaluations across multiple datasets demonstrate accurate tracking and high-quality rendering. The paper also shows the method's advantages in terms of runtime and storage.

3）	Figure 3 and Figure 4 effectively show the advantages of Splat-SLAM over other methods in rendering and reconstruction.

4）	The paper is well-written, clearly explaining the design rationale behind each component. For instance, the authors justify the choice of the DPT model for proxy depth estimation over newer alternatives.

**Weaknesses:**

1)  It would be better to include visualizations that illustrate loop closure results. For example, presenting complete reconstruction results of selected scenes and comparing these results across your method, ground truth, and other approaches would offer valuable insights.

2)  Minor:
Wrong numbers are highlighted in Table 1.  The section on Influence of Monocular Depth in the supplementary material is interesting, as it illustrates the upper bound of your approach. Since this content enhances the main paper, it may be worth considering moving this section to the main text.

**Questions:**

1)  Could you further explain how to construct proxy depth better in detail?

---

### Official Review · Reviewer_8LWN · 2024-10-31

**Soundness:** 2
**Presentation:** 3
**Contribution:** 1
**Rating:** 3
**Confidence:** 5

**Summary:**

The paper proposed an RGB-only SLAM system that utilizes a deformed 3D Gaussian representation for mapping, and a raft-based monocular tracking system with additional depth estimation. The system was evaluated on Replica, TUM, and ScanNet datasets, comparing with both NeRF-based and Gaussian-based SLAM systems, achieving superior or on-par performance with existing methods in tracking,
mapping and rendering accuracy while yielding small map sizes and fast runtimes.

**Strengths:**

- The paper conducted experiments on both synthetic and real-world datasets for tracking and mapping evaluation
- The paper writing is comprehensive and easy to understand
- The paper presents high performance among the baseline methods

**Weaknesses:**

- Lack of novelty. The paper significantly derives its methods from Droid-SLAM, especially in monocular tracking based on raft, with the addition of a depth estimation module previously introduced in Glorie-SLAM, for which appropriate credit is lacking in the methods section. Moreover, although the proposed deformed map offers a partial solution to BA-induced inconsistencies, it heavily relies on the existing monocular tracking method, which has limited contribution to addressing the core challenges of RGB-only SLAM (e.g. depth and scale estimation). Finally, the idea of loop-induced scene drift is not new, similar ideas have been proposed from LoopySLAM, LoopSplat etc.

- If the system employs a pre-existing tracking module for pose estimation and uses a Gaussian map for scene representation, it is essential to compare it with similar frameworks to highlight distinct advantages or improvements. For instance, Photo-SLAM (as mentioned in supplementary material) integrates ORB-SLAM3 for tracking, while RTG-SLAM incorporates ORB-SLAM2, both supporting monocular mode and utilizing 3DGS for map representation. This comparison is crucial for evaluating the proposed system within the context of existing solutions.

- Although the paper demonstrates high performance, the motivation for each proposed component is either derived from existing works or lacks clear connections. Given the focus of the ICLR conference on presenting novel insights to the community, this paper may not fully align with the conference's objectives.

References:
- Teed, Zachary, and Jia Deng. "Droid-slam: Deep visual slam for monocular, stereo, and rgb-d cameras." Advances in neural information processing systems 34 (2021): 16558-16569.
- Zhang, Ganlin, et al. "Glorie-slam: Globally optimized rgb-only implicit encoding point cloud slam." arXiv preprint arXiv:2403.19549 (2024).
- Liso, Lorenzo, et al. "Loopy-slam: Dense neural slam with loop closures." Proceedings of the IEEE/CVF Conference on Computer Vision and Pattern Recognition. 2024.
- Zhu, Liyuan, et al. "Loopsplat: Loop closure by registering 3d gaussian splats." arXiv preprint arXiv:2408.10154 (2024).
- Huang, Huajian, et al. "Photo-SLAM: Real-time Simultaneous Localization and Photorealistic Mapping for Monocular Stereo and RGB-D Cameras." Proceedings of the IEEE/CVF Conference on Computer Vision and Pattern Recognition. 2024.
- Peng, Zhexi, et al. "Rtg-slam: Real-time 3d reconstruction at scale using gaussian splatting." ACM SIGGRAPH 2024 Conference Papers. 2024.

**Questions:**

- How does the deformed map relate to RGB-only SLAM? Given that it utilizes depth information, this approach appears equally suitable for RGB-D SLAM. Addressing BA-induced map inconsistencies, as tackled by previous works like Loopy-SLAM and LoopSplat, is not unique to monocular systems. Therefore, the rationale for introducing a deformed map specifically within monocular SLAM is not convincingly articulated.

- Compared to LoopySLAM, which is based on NeRF, and LoopSplat, which uses Gaussian splatting, the primary distinction appears to be whether map deformation is applied per keyframe or per submap. Given the efficiency perspective, adjusting the submap seems more practical and feasible, as adjustments between adjacent keyframes are typically minimal. To validate this approach, further experiments focusing on efficiency and rendering quality are recommended.

- The paper employs metrics like PSNR, SSIM, and LPIPS to assess rendering quality. It remains unclear how these metrics are configured for evaluation—are they assessed at each keyframe during the SLAM system's runtime, post-mapping process, or on the reconstructed mesh? Further clarification on the evaluation setup for these metrics would enhance the understanding of their application and relevance.

- A typo on line 24,  RGB-only SLAM methods methods.

**Details Of Ethics Concerns:**

No ethics concerns.

---

### Official Review · Reviewer_Ckjz · 2024-11-06

**Soundness:** 3
**Presentation:** 3
**Contribution:** 2
**Rating:** 3
**Confidence:** 5

**Summary:**

The paper introduces a two-stage system to reconstruct globally consistent 3DGS. In the first stage, the system reconstructs camera poses and metric depth at each keyframe through dense bundle adjustment, where monocular depth estimation is leveraged for proxy depth fusion and additional constraints for dense BA. In the second stage, a globally consistent 3DGS is reconstructed in an incremental fashion where the 3DGS parameters will be updated once a new keyframe is added. To validate the effectiveness of the proposed method, the authors conducted experiments on existing public benchmarks and demonstrated better efficiency and performance over existing baselines in globally consistent 3DGS reconstruction.

**Strengths:**

1. The implementation of the system is sophisticated which requires extensive effort.
2. The utilization of monocular depth offers good regularization of depth map while the depth are still calculated per-pixel which maintains the accuracy.
3. The performance, especially the global consistency is superior to other global consistent 3DGS baselines.

**Weaknesses:**

1. The paper is more about a sophisticated system implementation for a specific problem and lacks some insights or contributions to the understanding of the problem. Thus, it is better to be published at other venues such as CVPR and 3DV.
2. The term "Deformable" is a little bit confusing since it is widely referred to deformable structure in computer vision literatures. Thus, it is better to replace this term with other word.
3. There are some missing but important details in Figure.2:
    - How the conv-gru facilitates the loop closure?
    - What is the flow revision? I believe it is refinement based on dense optical flow but it is better to be clearly indicated.
    - What is the \hat{D}, it should be the output of filtered depth, but should be included in one of the equations.
4. The DSPO is refereed as a "layer". However, it is only used in forward optimization in the paper and there is no backward backpropgation through it.

**Questions:**

How it performs on dataset that contains more forward camera motion such as TartanAir?
While 3DGS itself might perform poorly or fail, it is worth to study the camera pose itself and depth map.

---

### Note · Authors · 2024-11-17

I have read and agree with the venue's withdrawal policy on behalf of myself and my co-authors.